# Ultra-low threshold lasing through phase front engineering via a metallic circular aperture

Zhixin Wang [1✉], Filippos Kapsalidis [1], Ruijun Wang[1], Mattias Beck [1] & Jérôme Faist[1✉]

Semiconductor lasers with extremely low threshold power require a combination of small volume active region with high-quality-factor cavities. For ridge lasers with highly reflective coatings, an ultra-low threshold demands significantly suppressing the diffraction loss at the facets of the laser. Here, we demonstrate that introducing a subwavelength aperture in the metallic highly reflective coating of a laser can correct the phase front, thereby counter-intuitively enhancing both its modal reflectivity and transmissivity at the same time. Theoretical and experimental results manifest a decreasing in the mirror loss by over 40% and an increasing in the transmissivity by $10^4$. Implementing this method on a small-cavity quantum cascade laser, room-temperature continuous-wave lasing operation at 4.5 μm wavelength with an electrical consumption power of only 143 mW is achieved. Our work suggests possibilities for future portable applications and can be implemented in a broad range of optoelectronic systems.

[1] ETH Zürich, Institute of Quantum Electronics, Auguste-Piccard-Hof 1, Zürich 8093, Switzerland. ✉email: zhixwang@phys.ethz.ch; jfaist@ethz.ch

The mid-infrared frequency range is of great interest to molecular spectroscopy applications, as multiple greenhouse and pollutant gases have their fundamental absorption lines in this region[1,2]. Advances have led to sensitive mid-infrared detectors with below 1 nW Hz$^{-0.5}$ noise equivalent power[3], and mid-infrared chemical spectroscopy systems have been demonstrated with sub-milliwatt incident optical powers[4]. Given the fact that intersubband and interband cascade lasers have exhibited wall-plug efficiencies well above 10%[5,6], low threshold devices are prime candidates for portable sensing applications, potentially demanding the electrical operating power at the level of only tens of milliwatts. Reducing the threshold demands developing suitable laser cavities, with high quality-factor[7–11] and small volume[12–17].

The vertical cavity surface emitting laser[18] is a common choice for small volume and low dissipation devices, but, because of the unfavorable polarization selection rules for intersubband transitions, this is not a geometry that can be easily implemented for intersubband quantum cascade lasers (QCLs)[19]. Instead, Fig. 1a

illustrates another design which consists of narrow (around 1-μm wide) and short-ridge QCL active region fabricated with the buried-heterostructure process[20]. In such a design, the optical losses needed by room-temperature continuous-wave laser operation require high modal reflectivity of the ridge facets, ideally above 98%. Because of the high dielectric constants, noble metals, such as Au, serve as desirable candidates to realize highly-reflective coatings for semiconductor lasers, particularly when combined with low-refractive-index insulating materials[21]. These metallic coatings indeed significantly suppress the mirror loss. However, the modal reflectivity is still limited by the diffraction loss, as will be discussed in this paper. Mitigating the diffraction loss of such tightly confined structure requires engineering the near-field phase front below the wavelength scale.

Enabled by the development of nano-fabrication techniques, the electromagnetic field now can be controlled at the scales that are well below the optical wavelength[22–24]. This led to the discovery of a variety of unique physics, such as the extraordinary optical transmission[25] where Ebbesen et al. demonstrated that the

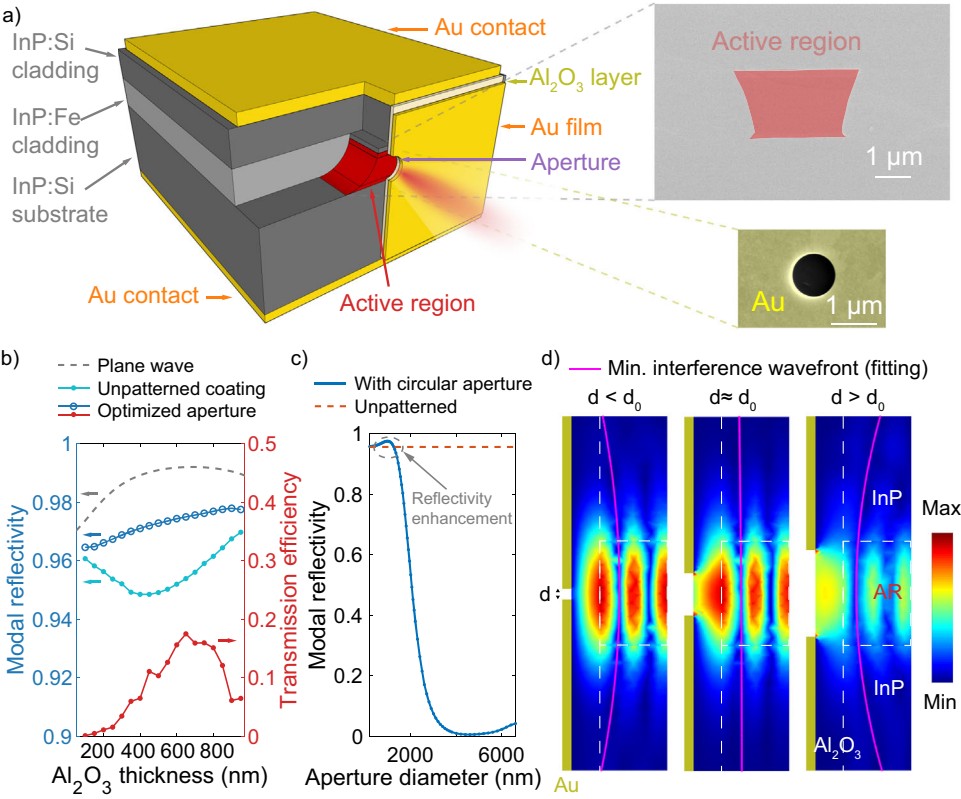

**Fig. 1 Laser design and theoretical results of the reflectivity enhancement. a** Illustrative image of the semiconductor laser structure where a hole is introduced in its front metallic coating. Here only half of the front coating is shown and the active region is drawn as exposed for a clear illustration. In the actual device, the active region is fully buried by the cladding and the coating fully covers the facet. An SEM photo of the laser facet before depositing the metallic coating is shown in the upper right inset. The lower right inset shows the SEM image of the highly reflective coating with a circular hole patterned by the focused ion beam milling. The aperture diameter is around 950 nm. The SEM images are colored manually. **b** Coating modal reflectivity simulated with varying thickness of the dielectric layer in three different cases, including the ideal plane-wave case (gray, dashed line), the case with a laser waveguide but without an aperture (light blue dotted line), and the case with a waveguide and a circular aperture through the metallic film of which the aperture size is optimized for each $Al_2O_3$ thickness (dark blue circled line). The red dotted line shows the transmission efficiency as a function of the $Al_2O_3$ thickness for the case with the circular aperture. The optical wavelength in the simulations is set as 4.5 μm. **c** Coating modal reflectivity with varying hole diameter, where the insulating layer thickness is set as $\lambda/4$. The straight dashed red line shows the reflectivity of the coating without an aperture. The dashed gray circle shows the region where the modal reflectivity is enhanced by the presence of the aperture. **d** Simulated field patterns of the coated laser facet using three different hole diameters. Shown in these plots are the amplitude of the vertical component of the electric field $|\mathbf{E}_z|$. Here the pink lines are the fitted curves for the minimum interference, showing the curvature of the optical wave front in each case. Parameter $d$ is the aperture diameter and $d_0$ is the diameter with the maximum modal reflectivity. In the three cases, the hole diameter is (from left to right): 250, 1000, 2000 nm, respectively. The color bar ranges in (**d**) are [$1.7 \times 10^4$, $2.5 \times 10^7$], [$2.1 \times 10^4$, $2.7 \times 10^7$], [$5.5 \times 10^4$, $4.0 \times 10^7$] (arbitrary units), respectively. © 2021 IEEE. Figure 1 is reprinted, with permission, from ref. [20].

normalized-to-area value of transmittance through two-dimensional (2D) periodic subwavelength apertures in a metallic film can be above unity. In such situations, the typical Kirchhoff analysis is no longer well-suited[26]. In 1944, Bethe presented a model where the diffracted waves are understood as caused by electromagnetic moments in the plane of the metallic hole[27]. A series of significant progress have been achieved afterward[28–37]. However, they mainly focused on the transmission enhancement[25,38] or the collimation of the transmitted light[22,39,40]. In this work, we do not focus on either of these, but instead demonstrate that the subwavelength aperture provides a way to alleviate the diffraction loss and enhance the modal reflectivity of the highly reflective metallic coating by engineering the near-field phase front of the beam.

## Results

**Theoretical simulations**. Under an ideal plane wave incidence, the facet reflectivity with a metallic coating as mentioned above is maximized with a quarter-wave-thick dielectric layer because the Ohmic absorption loss in the metal film is minimized. The dashed gray line in Fig. 1b shows the coating reflectivity dependence on the thickness of the insulating alumina ($Al_2O_3$) layer, computed with the consideration of a normal-incident 4.5-μm-wavelength plane wave. Assuming an Au film thickness of 200 nm, the reflectivity is maximized up to 99.2% with alumina that has the thickness of $\lambda/4$ (nearly 700 nm). The dotted light-blue curve in Fig. 1b shows the simulated modal reflectivity with a buried-heterostructure waveguide, where the lateral confinement is included. Because the beam propagating in the dielectric layer diverges, it cannot be perfectly coupled back into the waveguide after reflection, as shown in Supplemental Material Fig. S1a, resulting in the diffraction loss which increases with the dielectric layer thickness. As shown in Fig. 1b, the simulated reflectivity decreased to 95.6% when the diffraction loss is taken into account. Note that here the diffraction loss is higher than expected for a simple Gaussian beam approximation as the shape of the mode departs from the Gaussian. Moreover, because the transmissivity is ~$10^{-7}$, the light cannot be extracted through such a coating.

As a paradoxical result, the circled dark-blue curve in Fig. 1b indicates that both the modal reflectivity and the transmittance can be enhanced at the same time, by creating a circular subwavelength hole in the metal film aligned with the waveguide. Values of the aperture diameters at each dielectric thickness are shown in Supplemental Material Sec. A.2. The simulation indicates that with the same $\lambda/4$-thick alumina layer as mentioned above, the modal reflectivity of the facet can be enhanced to 97.6% by introducing a metallic hole that has a diameter of 990 nm. This means that the optical mirror loss is decreased by more than 40%. The ratio between the transmittance and the energy that is not reflected back ($T/(1 - R)$), defined in Fig. 1b as transmission efficiency, is shown as the dotted red curve. With $\lambda/4$-thick (700 nm) alumina, the transmission efficiency of the coating with a 990-nm-diameter metallic aperture is 15.9%. The transmissivity (0.4%) is enhanced by around $10^4$ compared to the case without any patterning, allowing the highly reflective coating to behave as a coupler. Figure 1c reports the simulated reflectivity at varying aperture sizes, where the thickness of the alumina is set as quarter-lambda (700 nm). As a reference, the orange dashed line indicates the modal reflectivity of an unpatterned coating. As highlighted by the gray circle, an increasing in the reflectivity is evidently seen in the subwavelength range. On the contrary, the reflectivity is dropped down to near zero when the diameter of the hole is comparable to the wavelength of the light. See Supplemental Material Sec. G for more details.

The paradoxical increase of the modal reflectivity with the metallic subwavelength hole can be understood in the view of a compensation in the retarded phase, following the proposal of Bethe. Because the diffraction field is interpreted as caused by the electromagnetic moments at the subwavelength metallic hole, these moments introduce additional phase. Such phase, with the optimized structural parameters, can compensate for the divergence of the light. As discussed in detail in Supplemental Material Sec. A.1, the aperture, in a reminiscence of the Arago spot[41], effectively refocuses the beam in the near-field as a lens would do, decreases the diffraction losses and therefore enhances the modal reflectivity. In order to show the influence of the aperture size on the optical field, the vertical electric field amplitude ($|E_z|$) of the coated waveguide facet is illustrated in Fig. 1d with varying hole diameters. Inside the laser, the reflected light interferes with the incident light, and the shape of the corresponding wave front can be depicted by the curvature of the intensity minimum, which is plotted as dark in Fig. 1d and highlighted by the pink fitting lines. In the case of a metallic hole with a small dimension, the intensity of the reflected field is almost constant. The modal reflectivity approaches maximum when the wave front gets flat, showing a suppression in the divergence. This is indicated by the middle image of Fig. 1d where $d \approx d_0 = 1$ μm. Here the flat shape of the wave front suggests a material compensation in the retarded phase of the diffracted field by the presence of the electromagnetic resonance of the metallic hole. As a comparison, the left and right images of Fig. 1d indicate that the phase front curves in negative or positive directions in the cases where the hole diameter is above or below the optimized value.

In this work, only circular-shaped aperture is considered, for the ease of fabrication tools. However, the shape of the aperture is not restricted. We show in Supplemental Material Sec. B that a higher transmission efficiency with nearly the same reflectivity can be achieved by an elliptical aperture.

**Experimental results**. Experimentally, metallic coatings are deposited on the buried-heterostructure QCL facets (see "Methods" section). The scanning electron microscope (SEM) photo of the laser facet before the coating is shown in the top right of Fig. 1a, indicating a small waveguide width. Using a procedure also described in the "Methods" section, we created a circular hole with the diameter of 950 nm in the Au film with a focused ion beam milling instrument. The SEM photo of the laser facet after the aperture milling is exhibited in the bottom right corner of Fig. 1a. For the benefit of aligning the hole to the laser ridge, auxiliary steps are introduced during the milling process, including marker etching and energy-dispersive X-ray (EDX) spectroscopy. See Supplementary Material Sec. C for more details.

In experiments, alumina/Au coatings are deposited to both sides of a 265-μm-long QCL. The alumina thickness is selected as 700 nm for an optimal balance in both the enhanced modal reflectivity and the transmission efficiency (Fig. 1b). The Au layer thickness is 200 nm. See Supplementary Material Sec. C for the SEM image of the whole device. The blue curves and dots in Fig. 2 report the continuous-wave power-voltage-current characterization of a coated QCL without patterning in the metal film. The waveguide width of the laser is around 2.5 μm. Here, a kink in the bias-current characteristics (solid blue curve) evidently shows a lasing threshold of 14.5 mA at room temperature. In the output power-current characteristics (dotted blue), the threshold is not identifiable because the extracted power is below the thermal-drift of our experimental environment.

Using the approach described in the "Methods" section, we then created a 950-nm circular hole in the metal film of the

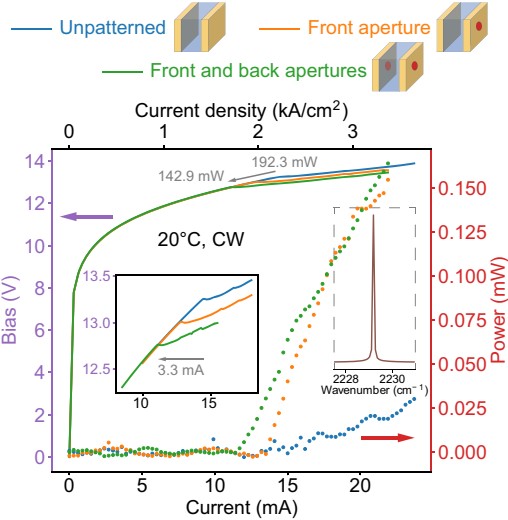

**Fig. 2 Lasing threshold reduction.** Continuous-wave (CW) current-voltage-power characteristics measured at room temperature with a short-cavity QCL where highly reflective metallic coatings are applied on both laser facets. The blue line and dots show the results where no apertures are patterned on the coatings. The orange line and dots show the results measured when a 950-nm-diameter circular hole is created in the metal layer of the coating on the front side. The green line and dots display the measured results when an additional 950-nm-diameter circular hole is milled in the Au film of the back-facet high-reflectivity coating. In these characterizations, the measuring step is 0.3 mA. A reduction of the threshold current is shown by the movement of the current-voltage kinks toward the left, and are more clearly displayed in the left inset, where the measurements are conducted using a finer current step of 0.1 mA. In total, the threshold dissipation power is reduced by over 25%. The laser is single-mode at the threshold, as shown in the right inset, where the spectral resolution of the measurement is 0.075 cm$^{-1}$. © 2021 IEEE. Figure 2 is reprinted, with permission, from ref. [20].

coating on the front facet of the QCL, in alignment to the laser ridge. The SEM photo is displayed in the lower-right corner of Fig. 1a. This aperture diameter is sufficiently close to the optimal point with inevitable errors. The orange data in Fig. 2 shows the measured power-voltage-current characterization results with the aperture in the front coating. As clearly indicated by the kink in the solid orange curve, the threshold current is now dropped to 12.9 mA, 11% lower than the one before the patterning. This is in agreement with our theoretical simulation which predicts 40% reduction in the mirror loss of the front facet. In addition, 150 μW emission power is detected through the front-side of the laser, commensurate with a transmission that increases by four orders of magnitude. This allows the lasing threshold to be also evident in the power-current characteristics (dotted orange in Fig. 2), although the value of threshold is more precisely shown by the bias-current kink.

Moreover, we replicated the same process of hole-milling on the back-facet Au film, and the threshold current drops to 11.2 mA. The corresponding characterization is presented as the green data in Fig. 2. Results with a smaller measurement step (0.1 mA) near the threshold are reported in the first inset of Fig. 2. The overall decreasing of the threshold is 3.3 mA. Figure 2 also shows a total reduction in the threshold consumption power by 25%. With optimized apertures in both facet-coatings, the threshold dissipation power at 20 °C is 143 mW. This is 45% smaller than the previously reported result[42]. The threshold current under −20 °C is 8.3 mA. In addition, the approach

presented in this work for decreasing the threshold can be well reproduced. See Supplemental Material Sec. D for the data measured with additional temperatures and devices.

The measured far-field patterns and the polarization characteristics of lasing beams are reported in Fig. 3a–c, which agree with the simulated results shown in Fig. 3d–i. The beam is mainly polarized in the vertical direction, agreeing with the selection rule of intersubband transitions[19]. Despite the circular symmetry of the aperture, the far-field pattern is highly asymmetrical and is much broader in the vertical direction. As shown by Fig. 3j, k which illustrates the near-field patterns, the electric field inside the aperture is oriented similarly as the one of a subwavelength dipole antenna[43]. As a result of the large diffraction angle, the vector Stratton-Chu model[44,45] must be applied for the far-field calculation and the scalar Kirchhoff theory[26,46,47] is no longer valid. As a result, the beam divergence in the vertical direction is much larger than the one along the horizontal direction. The presence of the metallic aperture strongly enhances the relative intensity of the far-field in the horizontal polarization created by the field discontinuity at the edges of the waveguide, as shown with more details in the Supplemental Material Sec. E.

## Discussion

In this work, we demonstrate that the mirror loss of a semiconductor laser can be substantially suppressed and the consumption power can be significantly decreased by controlling the optical near-field at the facets via creating a subwavelength metallic hole, leading to a mid-infrared QCL with the threshold dissipation power of only 143 mW in continuous wave at room temperature. We show that, in general, it is crucial to control the diffraction loss for achieving a low dissipation QCL. In the future, there is still room for further optimization. For example, using facets realized via dry etching techniques would enable much shorter cavities to be fabricated and consequently allow a lower threshold dissipation power while keeping the same reflectivity realized in this work. Moreover, the presented work could also potentially be applied in different frequency regions and a broader range of applications via modifying the dimension and geometry of the hole. See Supplemental Material Sec. F for more details.

## Methods

**Details on the fabrication process.** The active region of the QCLs presented in this work is based on a strained In$_{0.66}$Ga$_{0.34}$As/Al$_{0.665}$In$_{0.335}$As structure[48], grown by the method of molecular beam epitaxy (MBE). This active region incorporates two quantum cascade stacks centered at two different frequencies, 2325 cm$^{-1}$ (4.3 μm) and 2174 cm$^{-1}$ (4.6 μm), respectively. The laser is fabricated using the technique of buried-heterostructure process[49,50]. A first-order distributed-feedback (DFB) grating is etched to a depth of 100 nm in a 200-nm-thick InGaAs layer above the active region, with the grating period spread around a range of target wavelengths. The QCL is fabricated into a buried-heterostructure configuration. For the presented device shown in Figs. 2 and 3, the lattice constant and the filling factor of the grating are designed as 0.7327 μm and 54.59%, respectively. The lasing wavelength observed in the experiment (2229.2 cm$^{-1}$), which is shown by the inset of Fig. 2, is consistent with the Bragg wavelength of the DFB grating. The active region is patterned in ridges of 1–3 μm by wet etching. Metal-organic vapor phase epitaxy (MOVPE) is used for growing the claddings surrounding the active region. The cladding material on the sides of the active region is InP:Fe and the material on top is InP:Si. Ohmic contacts are deposited on both the top and the bottom sides of the device. The lasers are cleaved mechanically to the length of around 250 μm. Afterward, they are either mounted on aluminum nitride substrates with the epitaxial side down, or on copper mounts with the epitaxial side up. The metallic coatings, consisting of Al$_2$O$_3$ and Au layers, are grown by an electron beam evaporation process.

**Details on the focused ion beam milling.** The focused ion beam (FIB) milling technique is conducted using the Helios 5 UX (Ga ion) manufactured by Thermo-Fisher and maintained by ScopeM, ETH Zurich. For each facet, the FIB milling is used in two steps: before the deposition of the coating, alignment markers are milled by FIB on the laser facet about 20 μm away from the laser waveguide. After the coating

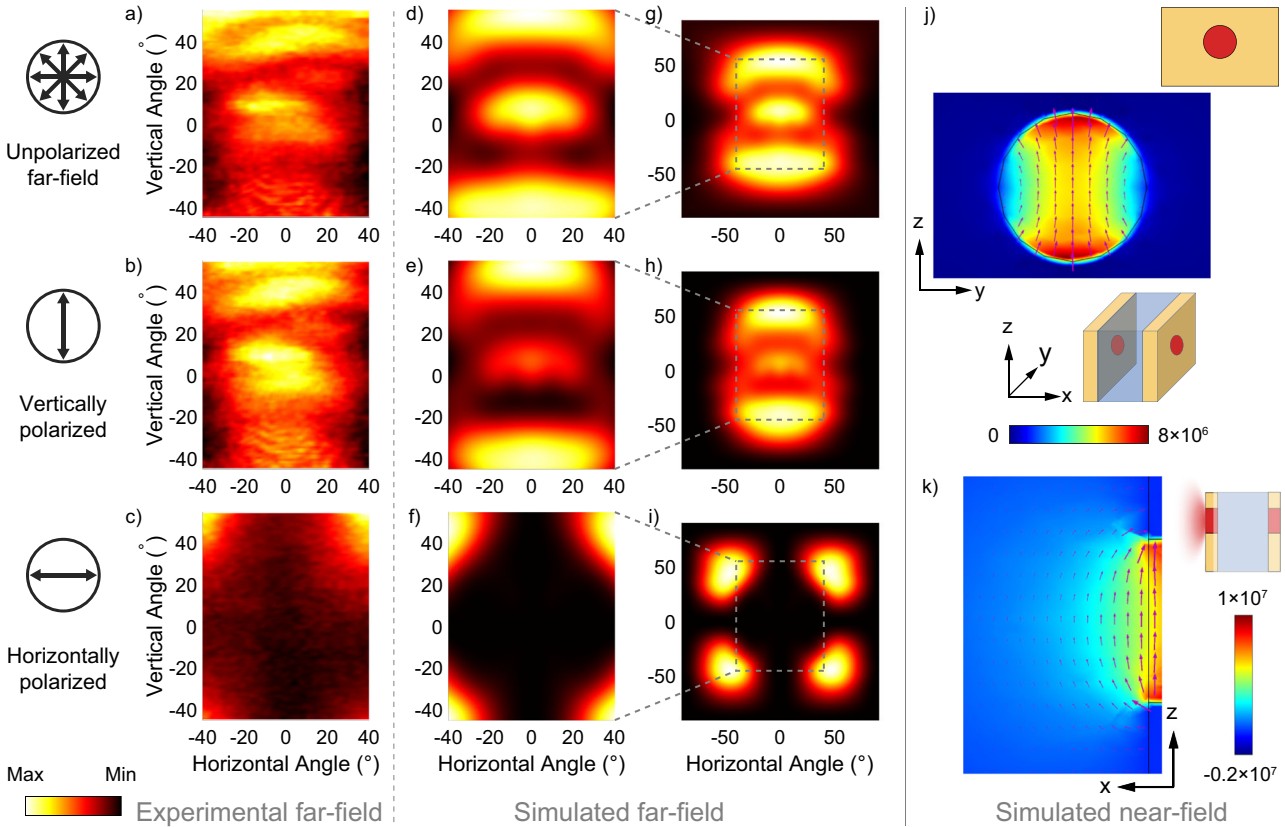

**Fig. 3 Far-field and near-field of radiation through the metallic aperture. a–i** Measured (**a–c**) and simulated (**d–i**) far-field patterns of the beam emitted through the circular aperture in the front metallic coating. **a, d, g** Unpolarized far-field beam patterns, **b, e, h** vertically polarized and **c, f, i** horizontally polarized. **a–c** Measured at room temperature with a bias voltage of 13.6 V. Plotted in (**a–i**) are the patterns of the energy ($|E|^2$). **j, k** Simulated near-field patterns near the circular aperture in the front view (**j**) and the side view (**k**). The colormap shows the vertical component of the electric field ($E_z$) and the vectors represent the electric field vectors. Upper right insets indicate the direction of the view in each plot. The plane of (**j**) is selected as the outer interface of the metallic coating and the plane of (**k**) is the cut-plane through the middle of the device. The color bar ranges in (**a–i**) are [3.2, 8.6], [2.5, 7.0], [0, 11], [0.3, 0.9], [0.3, 0.9], [0, 0.2], [0, 0.9], [0, 0.9], [0, 0.2] (arbitrary units), respectively.

deposition, the circular aperture is milled through the metallic layer with the help of the markers. The milling stops at the interface between the $Al_2O_3$ and Au layers. In the end, the alignment between the aperture and the waveguide is checked by a high-resolution scanning electron microscope (SEM) or energy-dispersive X-ray spectroscopy (EDX). Further details and images are shown in the Supplemental Material, Sec. C.

**Measurement devices**. For all experimental measurements, the laser is mounted on a Laboratory Laser Housing from Alpes Lasers. For the current-voltage-power characterization and the spectrum measurement, the laser is driven by a Keithley 2420 source-meter in continuous-wave operation. The current and voltage are measured by the same source-meter. The spectrum is measured with a Fourier-transform infrared (FTIR) spectrometer (Bruker, Vertex 80). For the far-field measurement, the laser is driven by an Agilent 8114A pulse generator. The far-field patterns are measured using a pyroelectric detector (Gentec-EO: THZ2I-BL-BNC) mounted on a motorized scanning stage. A DSP lock-in amplifier (Model 7265 by EG&G Instruments) is also used for the far-field measurement.

**Simulation details**. The simulated data shown in Figs. 1 and 3 are obtained with 3D COMSOL Multiphysics models. The active region is approximated as a rectangular-shaped waveguide with the width of 2.8 μm and the height of 2 μm. The cladding InP thickness is set as 3.37 μm. The boundary condition of the top cladding surface is set as perfect electric conductor (PEC). A numeric port is set to the entrance of the waveguide to excite the waveguide mode. Scattering boundary conditions are applied to all other boundaries, including the bottom substrate surface, the top surface of the coating layers and all the sidewalls. Due to the transverse mirror symmetry of the structure, the model is sometimes halved by the symmetry plane and perfect magnetic conductor (PMC) is applied on the cut-plane interface in order to reduce the number of the meshing elements without influencing the simulation results. The choice of PMC boundary condition is due to the TM polarization of the mode. The refractive indices of the $Al_2O_3$ and the evaporated Au are adopted from the built-in COMSOL library at the target wavelength of

4.5 μm. The refractive indices of InP and the active region are around 3.10 and 3.35, respectively.

## Data availability

The numerical simulation and experimental measurement data that support the plots within this paper are available from the corresponding author upon reasonable request. Data that support the findings of this article are also available in the ETH Research Collection, https://www.research-collection.ethz.ch/handle/20.500.11850/506863.

## Code availability

The code used in the current study are available from the corresponding author upon reasonable request.

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

## Acknowledgements

This research was partially funded through an unrestricted gift by Huawei Technologies. The authors would like to kindly thank the following people for their technical supports and fruitful discussions: Giacomo Scalari, Matthew Singleton, Bo Meng, Felice Appugliese, and Urban Senica from Institute of Quantum Electronics, ETH Zurich; Emilio Gini from FIRST, ETH Zurich; Joakim Reuteler and Peng Zeng from ScopeM, ETH Zurich; Chao Peng from Peking University.

## Author contributions

Z.W. and J.F. conceived the idea. F.K. and Z.W. fabricated the devices. Z.W. performed the theoretical simulations, experimental measurements, and data analysis. M.B. conducted the MBE active region growth. Z.W., F.K., R.W., and J.F. wrote the manuscript. J.F. supervised the research. All authors reviewed the manuscript.

## Competing interests

The authors declare no competing interests.
