## [Peer Review File · Nature Communications]

REVIEWER COMMENTS

Reviewer #1 (Remarks to the Author):

This study presents a novel route to achieving room temperature low threshold QCL lasing by milling a metallic circular aperture on the metal facet. The circular aperture plays two roles: providing a phase correction to improve reflectivity and allowing for outcoupling through extraordinary transmission phenomena. I think the results as presented are interesting, however the study as presented feels more like a preliminary result than a complete study. For a journal of this caliber, I would have expected more analysis. I think this manuscript could be reconsidered for this journal if more is done to analyze the underlying physics of what is going on. A few specific comments include the following:

- 1) The key idea of the work is low threshold lasing and low power consumption of QCLs. But there are already reports on low power consumption QCL at room temperature in Ref 34 and Nanoscale Research Letters 12, 281, 2017. The demonstrated threshold in this work is indeed a little lower but the novelty seems limited.
- 2) The authors empirically demonstrate that reflectivity is enhanced with the aperture but do not further elaborate. Going beyond Figure 1d, it would be helpful for them to further theoretically elucidate the mechanism of diffractive loss in the case of no aperture, how exactly the aperture plus reflector quantitatively functions as a parabolic reflector to combat diffraction, and the role that the index contrast between alumina and indium phosphide plays into this.
- 3) Following the prior point, the case of extraordinary transmission with guided modes/diffracted waves is very different from prior discussions of extraordinary transmission with incident plane waves. An angular spectrum decomposition of the diffracted wave and analysis of extraordinary transmission in this manner is an example of a more rigorous analysis than what is provided.
- 4) Can the alumina be further thinned down to minimize spurious diffraction? Are there other materials that could better mitigate the diffraction problem? In general, the alumina plays a major role in reflective losses and its precise role is not elaborated on beyond SI Figure A.
- 5) What about the shape of the aperture? Why be limited to just a circular opening? Given that these are ridge waveguides, what about a subwavelength-scale slit or set of circular apertures? Often we want the ability to tailor transmission based on application, it's not clear if this concept can generalize accordingly. Can polarization also be controlled?

Reviewer #2 (Remarks to the Author):

The manuscript by Wang et al. reports on the design, fabrication and testing of a mid-infrared quantum cascade laser (QCL), operating at a wavelength of 4.5 micrometers, with a record electrical dissipation at threshold as low as 143 mW. This achievement is about a factor of 2 better than previous state-of-the-art (ref. 34) and is obtained exploiting carefully optimized subwavelength openings in the metal coated front and back facets of a short cavity QCL. Such an improvement is ascribed to the enhancement of both the transmission efficiency and the modal reflectivity with respect to unpatterned coating.

The presented data and their interpretation are clearly described and valid. The main conclusion, that by engineering sub-wavelength patterns on the laser facets and particularly openings it is possible to significantly reduce optical cavity losses, is convincing.

On the other hand, the potential significance of the reported work is limited, being mainly incremental in nature and mostly restricted to the field of QCL optimization.

Indeed, the field of sub-wavelength patterning of QCL facets and wavefront engineering, even with sub-wavelength openings has started long ago (e.g. Yu et al. Opt. Exp 2008 DOI: 10.1364/OE.16.019447; Appl. Phys. Lett. 89, 093120 (2006); Nature Photonics volume 2, pages564–570 (2008); <https://doi.org/10.1364/JOSAB.27.000B18>)

Also, while the electrical power threshold reduction is surely interesting, the strong limitation in power output (0.1 mW), inherently related with the selected approach, in contrast with the manuscript claims, will probably limit potential open field or even hand-held applications. In fact, while coherent spectroscopies may use such a low power levels, other techniques more suitable for field applications (e.g. optoacoustic spectroscopy) will surely suffer from the limited power.

So, although I believe that the manuscript should be published, probably a more sectoral optical journal would be more appropriate.

The validity of the approach and the quality of the data is satisfactory. A list of minor notes are following.

- the authors claim that the proposed approach for wavefront engineering may be used for further applications like frequency comb dispersion compensation. Another generic claim is the possibility of tailoring the beam retarded phase. However, such important statements are too generic and not supported by the presented data.
- the authors should explain why they used a QCL emission region composed of a stack of 2 active regions characterized by different designed wavelengths. What is the advantage (if any) for the attainment of a low threshold?
- the observed emission wavelength (2229 cm^{-1}) is different from the designed wavelengths of the 2 active regions. The authors should clarify this point.
- the authors should add more details on how they managed to perform the facet coating milling on the back facet. From the photo in Fig. S3 the readers do not understand the structure of the laser holder and the actual accessibility of the back facet. Also, it may be useful to add details on the selective stopping of the milling on the Al₂O₃ layer.
- a color scale should be added to Fig. S2
- The photo in Fig. S3 c) does not show any DFB gratings. Also, more details on the DFB grating structure should be added.

Reviewer #3 (Remarks to the Author):

The authors presented a high-reflectivity coated mid-infrared QCL with simultaneously improved transmission and modal reflectivity. This was obtained by milling a sub-wavelength circular hole in the metallic coating and was explained in terms of phase compensation of the beam interacting with the aperture. Experimental evidence of the improved performances was provided by current-voltage-power characterization of four devices with and without the coating patterning while numerical simulations were used to study the effect of a change of the geometrical parameters and seem to agree with the experimental results. Overall, I find the solution adopted by the authors to be original towards the stated goal of reducing the power consumption of mid-infrared QCLs and the paper to be nicely compact. I have, however, a few questions and observations to make:

1) The authors focus on circular apertures. In their conclusion, they state that by tuning the shape and size of the aperture, further applications can be envisioned for the proposed approach. Did the authors consider using other shapes for the aperture in their current work? If so, can they provide some information to show what would be the result of a different shape on the reflectivity and

transmission?

2) In supplementary section D, the authors show the characterization of three other devices with the "same material, similar dimensions and coating structures" to prove the reproducibility of their approach. Indeed, the threshold reduction for all of them is a convincing proof. However, there are also significant differences in the extracted power, especially between device 1 and 3. This raises a question regarding how similar these devices are. In fact, I suggest quantifying the differences between these devices so that one can try to distinguish between the effect of different dimensions and different coating structures, for example.

3) The experiments reported in the main text were performed on a coated device with a Al₂O₃ thickness of 700nm. Yet, according to the simulations of fig. 1b, the maximum in the modal reflectivity is at 900nm, which would grant the greater threshold reduction. Therefore, one must conclude that this specific choice of Al₂O₃ thickness was done to maximize the transmission efficiency. The underlying criterium should be made clear in the text, though.

4) The use of noble metals for the fabrication of plain HR coatings for mid-infrared QCLs is described in the paragraph of line 27. Yet, this solution is not a new one. One can find previous instances in the literature, such as Page, H., P. Collot, A. Rossi, V. Ortiz and C. Sirtori. "High reflectivity metallic mirror coatings for mid-infrared ($\lambda \approx 9 \mu\text{m}$) unipolar semiconductor lasers." *Semiconductor Science and Technology* 17 (2002): 1312-1316 for example. A citation is consequently needed.

5) There are not many details about the simulations. As a bare minimum, I would like to know if the simulations were all performed in 3D and how the boundary conditions were set. Alternatively, the authors could add a paragraph specifying all relevant information.

Overall, I consider this work to be worth a publication in Nature Communications if the authors can address the above comments.

Response to the comments of reviewers

Zhixin Wang^{1,*}, Filippos Kapsalidis¹, Ruijun Wang¹, Mattias Beck¹, and Jérôme Faist^{1,*}

¹ETH Zürich, Institute of Quantum Electronics, Auguste-Piccard-Hof 1, Zürich 8093, Switzerland

*corresponding author: Zhixin Wang (zhixwang@phys.ethz.ch); Jérôme Faist (jfaist@ethz.ch)

1 Response to Reviewer 1

Major comment

This study presents a novel route to achieving room temperature low threshold QCL lasing by milling a metallic circular aperture on the metal facet. The circular aperture plays two roles: providing a phase correction to improve reflectivity and allowing for outcoupling through extraordinary transmission phenomena. I think the results as presented are interesting, however the study as presented feels more like a preliminary result than a complete study. For a journal of this caliber, I would have expected more analysis. I think this manuscript could be reconsidered for this journal if more is done to analyze the underlying physics of what is going on.

Answer

Thank you for the comment. Considering the comment about the preliminary nature of the study, we agree that the previous version of the manuscript did not fully reflect the extensive studies we made. We made substantial changes in the revised manuscript. Mainly, we updated the following parts with detailed explanation and supporting data:

- Supplemental Material, Section A. “Additional data and explanation of the reflectivity enhancement”, new subsection A.1 “Beam focusing by the metallic aperture”:

To complement the explanation of phase front correction mentioned in the main text, we investigated how the beam shape changes with varying dielectric layer thickness, aperture dimension and during the propagation. We show in a detailed way that the metallic aperture behaves effectively as a lens and refocuses the reflected beam. This focusing effect mainly occurs in the horizontal direction due to a change in the boundary conditions. Such refocusing improves the coupling efficiency between the back-reflected beam and the laser waveguide, reduces the diffraction loss and enhances the modal reflectivity. Focusing the beam is physically equivalent to flattening the phase front.

- Supplemental Material, new Section B. “Analysis on the aperture shape”:

We analyze the reflectivity and the transmissivity dependence on the shape of the aperture using 3D COMSOL simulation. We show that compared to the circular aperture as presented in the main text, a higher transmission efficiency with nearly the same reflectivity can be achieved by an elliptical aperture. We also discover that the optimization of the modal reflectivity is more sensitive to the width of the aperture than the height, in agreement with the polarization feature of the structure as discussed in the manuscript.

- Supplemental Material, Section C “Additional information on the focused ion beam milling”

We show the workflow of the aperture milling with corresponding images. We additionally show the side-view photos of the vacuum chamber while conducting the FIB milling on the front and the back facets of the lasers. We elaborated more details on the FIB milling.

- Supplemental Material, new Section F. “Reflectivity enhancement in TE modes at the wavelength of 1550 nm”

We show that the presented reflectivity enhancement effect is not limited to TM polarized QCL devices. As an example, 35% reduction in the mirror loss is presented with a simulation for a metallic coating of a laser at the wavelength of 1550 nm under TE polarization.

- Supplemental Material, new Section G. “Transmission enhancement at a large aperture dimension”:

Although not within our major scope of interest, we investigated the transmission enhancement of a metallic aperture at a large aperture diameter using 3D COMSOL simulations. Above-unity normalized-to-area transmission is observed, although it is not the extraordinary optical transmission due to the limited enhancement. The parameters of the transmission enhancement are far away from the parameters used in our experiments where the reflectivity enhancement is demonstrated.

- Main text, Section “Methods”, new subsection “Simulation details”

We show the details of the 3D COMSOL simulations conducted for the main text.

With the listed substantial changes and unlisted updates, we believe the manuscript becomes much more solid and complete than the first version. We also believe that the underlying physics is analyzed and explained in a much more clear and detailed way.

Specific comment 1

The key idea of the work is low threshold lasing and low power consumption of QCLs. But there are already reports on low power consumption QCL at room temperature in Ref 34 and *Nanoscale Research Letters* 12, 281, 2017. The demonstrated threshold in this work is indeed a little lower but the novelty seems limited.

Answer

Indeed, the presented work is not the only one towards low dissipation QCLs. However, the impact and novelty of this work is far beyond the record of dissipation power and the topic of QCLs. In our opinion, the most crucial point in this work is the paradoxical phenomenon that both the modal reflectivity and the transmission can be simultaneously improved rather than merely a new record in the QCL threshold dissipation. The underlying physics of the presented phenomenon, i.e. the electromagnetic dipole resonance of a subwavelength metallic aperture, is interesting for the whole community. As mentioned in the revised manuscript, this effect is not limited to QCL devices. For example, simulated results of modal reflectivity enhancement on a TE-polarized structure at the wavelength of 1550 nm is shown in the Supplemental Material Sec. F.

We believe that the counter-intuitive nature and possible applications of our presented work are exciting. The presented approach to enhance the coating reflectivity is effective and technically simple, but not at all trivial.

Specific comment 2

The authors empirically demonstrate that reflectivity is enhanced with the aperture but do not further elaborate. Going beyond Figure 1d, it would be helpful for them to further theoretically elucidate the mechanism of diffractive loss in the case of no aperture, how exactly the aperture plus reflector quantitatively functions as a parabolic reflector to combat diffraction, and the role that the index contrast between alumina and indium phosphide plays into this.

Answer

Elucidating the mechanism of diffraction loss

Thank you for the comment. As schematically illustrated in Fig. S1(a) of the Supplemental Material, the beam diverges inside the dielectric layer because the lateral confinement is lifted. When the diverging beam finally arrives at the laser facet, it becomes much broader and cannot be perfectly coupled back into the waveguide. Thus the diffraction loss arises.

As shown in Figs. S3 (a,b) and S4 of the Supplemental Material, the reflected beam is refocused by the optimized metallic aperture. The focusing occurs mainly in the horizontal direction due to the change in the boundary conditions. Figures S3 and

S5 show that this focusing effect vanishes if the parameters of the structure (aperture dimension and dielectric layer thickness) are tuned away from the optimized points.

We agree that the physical mechanism was not elucidated clear enough, and we revised our manuscript accordingly as follows:

- Line 30: *“Under an ideal plane wave incidence, the coating reflectivity is maximized with a quarter-wave-thick dielectric layer because the Ohmic absorption loss in the metal film is minimized.”*
- Line 35: *“Because the beam propagating in the dielectric layer diverges, it cannot be perfectly coupled back into the waveguide after reflection, as shown in Supplemental Material Fig. S1(a), resulting in the diffraction loss which increases with the dielectric layer thickness. In fact, with the diffraction losses taken into account, the computed reflectivity drops to 95.6%. Note that here the diffraction loss is higher than expected for a simple Gaussian beam approximation as the shape of the mode departs from the Gaussian. ”*
- Line 57: *“As discussed in details in Supplemental Material Sec. A.1, the aperture, in a reminiscence of the Arago spot, effectively refocuses the beam in the near-field as a lens would do, decreases the diffraction losses and therefore enhances the modal reflectivity.”*
- Supplemental Material, new subsection A.1 “Beam focusing by the metallic aperture” in the updated Section A. “Additional data and explanation of the reflectivity enhancement”

Index contrast between alumina and the waveguide

As mentioned in the answer to Specific comment 4 of Reviewer 1, one advantage of Al_2O_3 is that the index contrast to the InP waveguide is large, resulting in a high reflection at the interface between the waveguide and the dielectric layer, which benefits a high modal reflectivity of the coating. However, the physics of the presented work is generally valid and selection of the dielectric material is not exclusive to Al_2O_3 .

Specific comment 3

Following the prior point, the case of extraordinary transmission with guided modes/diffracted waves is very different from prior discussions of extraordinary transmission with incident plane waves. An angular spectrum decomposition of the diffracted wave and analysis of extraordinary transmission in this manner is an example of a more rigorous analysis than what is provided.

Answer

We emphasize that in this work, we focus on the enhancement in the modal reflectivity and the consequent control of the diffraction loss. Transmission enhancement through a metallic aperture does NOT align with our major scope of interest.

In spite of that, transmission enhancement (above unity normalized-to-area transmissivity) is observed in our 3D COMSOL simulation at both two cases: under plane-wave incidence, and under waveguide mode incidence, as explained in the Supplemental Material Sec. G. In the case of the waveguide mode incidence, the maximum normalized-to-area transmissivity is 1.16 when the aperture diameter is near 3000 nm. Although it is above one, the value of enhancement is limited. Therefore, we clarify that this is NOT extraordinary optical transmission, but rather only transmission enhancement. In addition, the parameters are far from our experimental ones and it is not our focusing point in this work.

We agree that the original text was unclear and we updated it as follows:

- Line 14: *“Substantial progress in both theoretical and experimental aspects has been made since then, but mainly focused on the transmission enhancement or the collimation of the transmitted light . In this work, we do not focus on either of these, but instead demonstrate that subwavelength apertures provide a way to engineer the reflectivity of the metal film by tailoring the retarded phases of the beam. ”*

- Line 53: *“In contrast, a minimum in reflectivity, which is close to zero, appears as the aperture diameter becomes comparable to the laser wavelength. See Supplemental Material Sec. G for more details.”*
- Supplemental Material, new Section G. *“Transmission enhancement at large aperture dimension”*

Specific comment 4

Can the alumina be further thinned down to minimize spurious diffraction? Are there other materials that could better mitigate the diffraction problem? In general, the alumina plays a major role in reflective losses and its precise role is not elaborated on beyond SI Figure A.

Answer

Decreasing the dielectric layer thickness indeed reduces the diffraction loss, but also increases the Ohmic absorption loss by the metal film. As shown by the gray dashed line (plane wave) and the dark-blue circled line (coating with optimized aperture) in Fig. 1(b) of the main text, the modal reflectivity is actually worse if we continue decreasing the dielectric layer thickness from the selected quarter-wave value (700 nm).

There are several roles played by the Al_2O_3 . Firstly, as an insulating material, it prevents the laser from being short-circuited by the metal film. Secondly, its negligible absorption at $4.5\ \mu\text{m}$ satisfies the need for maximizing the modal reflectivity. Thirdly, the low refractive index (1.6) compared to InP (3.1) and the active region (3.4) maintains a high reflectivity at the interface between the laser waveguide and the dielectric layer. In principle, any insulating material that is transparent at the target wavelength with a low refractive index can be used as the dielectric layer in the metallic coatings. Al_2O_3 is a standard choice in the mid-infrared as a stable oxide with good transmission and adhesion properties.

The Al_2O_3 layer thickness has to be carefully chosen for enhancing the performance of the metallic coating, as mentioned in the following sentences of the manuscript:

- Line 30: *“Under an ideal plane wave incidence, the coating reflectivity is maximized with a quarter-wave-thick dielectric layer because the Ohmic absorption loss in the metal film is minimized”*
- Line 78: *“The Al_2O_3 thickness is selected as 700 nm for an optimal balance in both the enhanced modal reflectivity and the transmission efficiency [Fig. 1(b)]. The Au layer thickness is 200 nm.”*
- Supplemental Material, Sec. A.1, line 61: *“We further analyzed the dependence of the beam shape at the target plane on the dielectric layer thickness L , as plotted in Fig. S5. ... Figure S5 shows that when L is too small, two off-center lobes are clearly observed along the vertical symmetry axis of the beam. For the aperture diameter $D = 950\ \text{nm}$, the beam shape becomes single-lobed with L near or larger than 700 nm. On the other hand, the beam gets much broader if L is way larger than the 700 nm. Therefore, the aperture diameter must be matched with the dielectric layer thickness in order to achieve the optimal reflectivity, as shown in Fig. S6.”*

Specific comment 5

What about the shape of the aperture? Why be limited to just a circular opening? Given that these are ridge waveguides, what about a subwavelength-scale slit or set of circular apertures? Often we want the ability to tailor transmission based on application, it's not clear if this concept can generalize accordingly. Can polarization also be controlled?

Answer

Single aperture shape

Thank you for the comment. The shape of the aperture is not limited to the circular one, but was chosen for its relative ease of the fabrication, as written in the updated manuscript:

- Line 67: *“In this work, only circular shaped aperture is considered, for the ease of fabrication tools. However, the shape of the aperture is not restricted. We show in Supplemental Material Sec. B that a higher transmission efficiency with nearly the same reflectivity can be achieved by an elliptical aperture.”*
- Supplemental Material Sec. B, line 84: *“In the main text, we focus on the circular aperture shape because our experimental tool for the FIB milling, the Helios 5 UX, prefers a circular pattern to elliptical shapes. But in principle, the choice of circular shape is not mandatory. We also analyzed other aperture shapes with 3D COMSOL simulation. Because the waveguide shape is horizontally symmetric and quasi-symmetric in the vertical direction, the explored aperture shape is set as ellipses, which are symmetrical along both vertical and horizontal directions.”*

We added an additional section, Sec. B “Analysis on the aperture shape” into the Supplemental Material of the manuscript to describe the dependence of the reflectivity enhancement effect on the aperture shape. In this section, we show that a higher transmission efficiency with nearly the same reflectivity can be achieved by an elliptical aperture. Near the optimal reflectivity region, the modal reflectivity and the transmissivity is more sensitive to the width of the aperture than the height of it, agreeing with the vectoral nature of the aperture resonance radiation discussed in the main text (line 104: *“As a result of the large diffraction angle, the vector Stratton-Chu model must be applied for the far-field calculation and the scalar Kirchhoff theory is no longer valid”*).

In the main text of the manuscript, we focus on the circular aperture shape due to the preference of our FIB milling tool, the Helios 5UX. The circular aperture is almost as good as the elliptical one but much easier to be manufactured, as written in the updated manuscript:

- Supplemental Material, Sec. B, line 95: *“The difference between the reflectivity with the circular aperture (red star) and the maximum one with the elliptical aperture (800 nm high, 1000 nm wide) is less than 0.05%. The transmission efficiency of the circular aperture with 950 nm diameter is lower than the one of the elliptical aperture (800 nm high, 1000 nm wide) by 3%. Therefore, the circular aperture is almost as good as the elliptical one but much easier to be manufactured.”*

Aperture array

In principle, the patterning is not limited to a single aperture. An array of aperture would also perform the near-field phase correction. However, being remarkably simple in techniques is one major advantage of our presented approach to mitigate the diffraction loss on a metallic coating. In the manuscript, we have demonstrated and confirmed that one single aperture is able to significantly enhance both the modal reflectivity and the transmission at the same time. Using array of apertures would require large additional effort in both the design optimization and the experimental realization. Nevertheless, extending the solution from a single aperture to an array of apertures remains a feasible and promising outlook of this work in the future.

Speaking of aperture arrays, we here emphasize that the underlying physics of our presented work is fundamentally different from plasmonic antenna-array collimators published by Nanfang Yu, et al.¹. In this work, we are not targeting at collimating the far-field beam but instead reflectivity enhancement, and surface plasmonic effect is not the major reason for that. Our patterning does not contain surrounding metal grating structures but only requires a single subwavelength aperture. More discussions on the our differences can be found in the answer to the Major comment by Reviewer 2.

Polarization control

In this work, the major application scenario is focused on reducing the threshold dissipation and improving the performance of our laser. Polarization control of the transmitted beam is not within our major scope of interest. Nevertheless, the ratio of the horizontal polarization in the far-field beam is indeed increased by the presence of the metallic aperture, as discussed in the main text and the Supplemental Material Sec. E “Additional details of the far-field and near-field patterns”:

- Line 107: *“The presence of the metallic aperture strongly enhances the relative intensity of the far field in the horizontal polarization created by the field discontinuity at the edges of the waveguide, as shown with more details in the*

Supplemental Material Sec. E.”

- Supplemental Material, Sec. E, line 150: “*For the uncoated laser facet, COMSOL simulation shows that the peak power ratio between the horizontally polarized far-field beam and the vertically polarized one is 4%. In the case of the metallic coating with a 950 nm-diameter aperture as presented, this ratio is increased to 24%, showing that the relative intensity of the far field in the horizontal polarization is significantly enhanced by the presence of the metallic aperture, as mentioned in the main text.*”

2 Response to Reviewer 2

Major comment

... On the other hand, the potential significance of the reported work is limited, being mainly incremental in nature and mostly restricted to the field of QCL optimization. Indeed, the field of sub wavelength patterning of QCL facets and wavefront engineering, even with sub-wavelength openings has started long ago (e.g. Yu et al. Opt. Exp 2008 DOI: 10.1364/OE.16.019447; Appl. Phys. Lett. 89, 093120 (2006); Nature Photonics volume 2, pages564–570 (2008); <https://doi.org/10.1364/JOSAB.27.000B18>). Also, while the electrical power threshold reduction is surely interesting, the strong limitation in power output (0.1 mW), inherently related with the selected approach, in contrast with the manuscript claims, will probably limit potential open field or even hand-held applications. In fact, while coherent spectroscopies may use such a low power levels, other techniques more suitable for field applications (e.g. optoacoustic spectroscopy) will surely suffer from the limited power.

Answer

Comparison with previous works

Thank you for the comment. We agree that our manuscript was not clearly stating the difference with previous work, this fact is now mentioned in the updated introduction (line 14, “*Substantial progress in both theoretical and experimental aspects has been made since then, but mainly focused on the transmission enhancement or the collimation of the transmitted light . In this work, we do not focus on either of these, but instead demonstrate that subwavelength apertures provide a way to engineer the reflectivity of the metal film by tailoring the retarded phases of the beam.*”). We are aware that subwavelength openings were performed previously and the mentioned publication on Nature Photonics² was cited in the initial manuscript. However, here we emphasize that the goals and the underlying physics between our work and the mentioned references are essentially different. The major differences are as follows:

1. Most of the mentioned works are aiming at collimating light after extraction, not at changing the reflectivity of the coating. Neither of the previous work demonstrates reflectivity enhancement or threshold reduction by the metallic aperture. Actually, in Fig. 5 of the mentioned Ref.², it is shown that the threshold of the QCL is increased because of the patterning in the metallic coating. On the contrary, the major goal of patterning the metallic coating in our work is to control the diffraction loss and to engineer the modal reflectivity. The difference in the goals of the works is the most crucial.
2. For the need of beam collimation, plasmonic properties are required in the mentioned references. This is however not the major factor in our work. As discussed in the reference mentioned by the reviewer³, the effect of the surface plasmons are dependent on the thickness of the metal layer. This is one of the reasons that in one of their mentioned publications², the Au layer is as thick as 1.7 μm in the design where the only the coating is sculpted and the semiconductor interface remains flat (Fig. 1b in Ref.²). On the contrary, we show in Fig. R1 that the modal reflectivity of the our perforated coating is barely influenced by the Au layer thickness. The difference in the reflectivity between a 300 nm-thick patterned coating and a 100 nm-thick one is less than 0.3%, which is minor compared to the nearly 2% change induced by the aperture opening as discussed in the main text.

Figure R1. Modal reflectivity as a function of the Au layer thickness, obtained by 3D COMSOL simulation. The Al_2O_3 layer thickness is 700 nm and the diameter of the metallic aperture is 950 nm. The y-axis is in the same range as Fig. 1(b) in the main text.

3. There is no groove gratings or arrays involved in our work, which is commonly applied in the mentioned publications. The metallic coating in our work is only patterned by one single aperture.
4. In Ref.², the dielectric layer is as thin as 200 nm and the major role of it is to electrically insulate the semiconductor waveguide from the metallic film. In our work, the dielectric layer thickness is also optimized for controlling the optical losses, as shown in Fig. 1(b) of the main text and explained in the answer to the Specific comment 4 of Reviewer 1.

We additionally cited most of the mention publications in the revised manuscript.

Power output

We believe that the output optical power of the presented device does not limit the impact of this work. In fact, infrared detectors that are capable of measuring infrared radiations as little as a fraction of a nW have already been commercialized⁴. For the DLaTGS detector which is commonly used in the mid-infrared, the noise equivalent power (NEP) can be tens of nW with a sub-millimeter element size. Therefore, such low dissipation lasers would be very useful for spectroscopy applications.

Moreover, as answered to the Specific comment 1 by Reviewer 1, the most crucial point of this work is the paradoxical phenomenon that both the modal reflectivity and the transmission can be simultaneously improved. Although a clearly better record in the threshold dissipation is demonstrated, we believe that the impact and the physical novelty of this work is far beyond the values of laser performances.

Specific comment 1

the authors claim that the proposed approach for wavefront engineering may be used for further applications like frequency comb dispersion compensation. Another generic claim is the possibility of tailoring the beam retarded phase. However, such important statements are too generic and not supported by the presented data.

Answer

Thank you for the comments. We agree that the use of such an aperture for dispersion compensation would require completely new optimizations and discussions, therefore we no longer mention “frequency comb dispersion compensation” in the revised manuscript. The manuscript is updated as follows:

- Last line in the “Abstract” section: *“Our work suggests possibilities for further applications and can be implemented in a broad range of optoelectronic systems”*
- Line 114: *“Furthermore, by tuning the size and the shape of the aperture, this work could be further extended to more applications and other wavelength ranges. See Supplemental Material Sec. F for more details.”*

Tailoring the beam phase

As explained in the revised manuscript, the phase front of the diverging beam inside the dielectric layer of the coating is influenced by the electromagnetic resonance of the metallic aperture. With the optimized parameters, such phase changing leads to a flatter wave-front and a lower diffraction loss. In this way, the beam phase is tailored. Correcting the diverging phase front is physically equivalent to effectively focusing the diverging beam. They both result in a lower diffraction loss and a higher modal reflectivity.

We revised our manuscript as follows:

- Line 35: *“Because the beam propagating in the dielectric layer diverges, it cannot be perfectly coupled back into the waveguide after reflection, as shown in Supplemental Material Fig. S1(a), resulting in the diffraction loss which increases with the dielectric layer thickness. In fact, with the diffraction losses taken into account, the computed reflectivity drops to 95.6%. Note that here the diffraction loss is higher than expected for a simple Gaussian beam approximation as the shape of the mode departs from the Gaussian.”*
- Line 57: *“As discussed in details in Supplemental Material Sec. A.1, the aperture, in a reminiscence of the Arago spot, effectively refocuses the beam in the near-field as a lens would do, decreases the diffraction losses and therefore enhances the modal reflectivity.”*
- Supplemental Material, new subsection A.1 “Beam focusing by the metallic aperture” in the updated Section A. “Additional data and explanation of the reflectivity enhancement”

In this part, we investigated how the beam shape changes with varying dielectric layer thickness, aperture dimension and during the propagation. We show in a detailed way that the metallic aperture behaves effectively as a lens and refocuses the reflected beam. This focusing effect mainly occurs in the horizontal direction due to a change in the boundary conditions. Such refocusing improves the coupling efficiency between the back-reflected beam and the laser waveguide, reduces the diffraction loss and enhances the modal reflectivity. Focusing the beam is physically equivalent to flattening the phase front.

Specific comment 2

the authors should explain why they used a QCL emission region composed of a stack of 2 active regions characterized by different designed wavelengths. What is the advantage (if any) for the attainment of a low threshold?

Answer

Thank you for the comment. The dual stack active region has a broad gain in the spectral domain, and enables devices to lase over a wide frequency range, which could be achieved by even shorter cavity lengths. This epilayer was therefore selected for the device fabrication of this work.

In general, the dual stack design does not lead to a lower lasing threshold. The threshold current density of the dual stack active region used here is estimated to be slightly higher than the single stack one, by no more than 10-20%.

The major novelty of this work, i.e. the simultaneous enhancement in both the modal reflectivity and the transmission by a metallic aperture, is not exclusive to a certain choice of the active region designs.

Specific comment 3

the observed emission wavelength (2229 cm⁻¹) is different from the designed wavelengths of the 2 active regions. The authors should clarify this point.

Answer

Thank you for the comment. This is due to the mode selection of the DFB grating. We agree that the lasing wavelength and the details of the DFB grating was not clarified in the initial manuscript, and we revised the manuscript as follows:

- Line 123: *“The grating is etched to a depth of 100 nm in a 200 nm thick InGaAs layer above the active region. For the presented device shown in Figs. 2, 3, the lattice constant and the filling factor of the grating are designed as 0.7327 μm and 54.59%, respectively. The lasing wavelength observed in the experiment (2229.2 cm⁻¹), which is shown by the inset of Fig. 2, is consistent with the Bragg wavelength of the DFB grating.”*

Specific comment 4

the authors should add more details on how they managed to perform the facet coating milling on the back facet. From the photo in Fig. S3 the readers do not understand the structure of the laser holder and the actual accessibility of the back facet. Also, it may be useful to add details on the selective stopping of the milling on the Al₂O₃ layer.

Answer

Thank you for the comment. We agree that we should add more details on the FIB milling process, and we rewrote the Supplemental Material Sec. C “Additional information on the focused ion beam milling” with the following major changes:

- Fig. S9
Workflow of the patterned metallic coating on a processed laser. Corresponding images at different steps are shown on the right side.
- Fig. S10
Side-view images of the vacuum chamber when conducting the FIB milling on the front (a) and back (b) facets of the coated lasers.
- Supplemental Material, Sec. C, line 110:
“Start from a mounted device. At least two sets of markers are milled by FIB on the semiconductor facets around 20 μm away from the active region waveguide for relocating the waveguide after coating deposition. Metallic coatings consisting a sequence of Al₂O₃ and Au layers are deposited via electron beam evaporation on both the front and the back facets of the laser. The deposition rate needs to be calibrated in order to control the thicknesses, especially for the dielectric layer. Circular patterns are milled in the front and the back facets with FIB milling. After that, the alignment between the aperture and the waveguide can be checked with the energy-dispersive X-ray (EDX) spectroscopy as shown in the bottom right image of Fig. S9. In order to penetrate the thick coating layers, high operating voltages and currents are needed.
“While performing the FIB milling, we also take real-time SEM and FIB images to monitor the milling progress. In this way, we are able to stop the milling immediately as soon the Au layer is fully patterned through. The milling rate of Al₂O₃ is much smaller than the one of Au given the same conditions. In experiments, the Au layer thickness is around or less than 200 nm. A circular aperture with 1 μm diameter through such a film requires only few seconds at a FIB

operating current of 26 pA and the voltage of 30 kV. Because we are monitoring the milling process in real time and the milling rate difference between Al₂O₃ and Au is huge, the milling into the Al₂O₃ layer is negligible, if there is any.

“Figures S10(a,b) show the side-view photos of the vacuum chamber when conducting the FIB milling on the front (a) and back (b) facet coatings. The electron beam gun is in the vertical direction (90°) and the ion beam gun is at the oblique direction (≈ 52°). The operating facet of the sample should be rotated to the direction normal to the ion beam gun for the FIB milling.”

“Performing the FIB milling on the back facet is essentially the same as doing it on the front facet, both of which are described by the workflow shown in Fig. S9. The only difference is that when operating the back facet milling, the device is placed on a customized 45° holder and the orientation of the device is flipped such that the back facet is facing the ion beam gun, as shown in Fig. S10(b).”

Specific comment 5

a color scale should be added to Fig. S2

Answer

Thank you for the comment. This figure is updated accordingly in the revised manuscript.

Specific comment 6

The photo in Fig. S3 c) does not show any DFB gratings. Also, more details on the DFB grating structure should be added.

Answer

Thank you for the comment. In our design, the active region is fully surrounded by the InP. Therefore, in a cross-sectional image of the laser facet which is perpendicular to the waveguide direction, one should only see an active region surrounded by the InP material, regardless of whether the cross-section is within the grating grooves or the grating bars. If the cleaving happens to be within the low-index groove of the grating, the active region in the SEM image will be 100 nm thinner than the case where the cleaving happens to be on the high-index bars of the grating. The grating structure is expected to be observed in the image of a longitudinal cut plane, not the transverse one.

Fig. S3(c) in the initial manuscript (bottom image of Fig. S9 in the revised manuscript) is the EDX image of the coated facet. The laser facet is covered by 700 nm thick Al₂O₃ and 200 nm thick Au. Because of the reasons mentioned above, it is natural that we do not see the 100 nm-thick grating in Fig. S9.

The bottom figure in Fig. S9 shows the alignment between the aperture and the laser waveguide in a clear and direct way, as written in the manuscript (Supplemental Material, line 114):

“After that, the alignment between the aperture and the waveguide can be checked with the energy-dispersive X-ray (EDX) spectroscopy as shown in the bottom right image of Fig. S9. In order to penetrate the thick coating layers, high operating voltages and currents are needed.”

We agree that the details of the DFB grating was not clarified in the initial manuscript, and we revised the manuscript as follows:

- Line 123: *“The grating is etched to a depth of 100 nm in a 200 nm thick InGaAs layer above the active region. For the presented device shown in Figs. 2, 3, the lattice constant and the filling factor of the grating are designed as 0.7327 μm and 54.59%, respectively. The lasing wavelength observed in the experiment (2229.2 cm⁻¹), which is shown by the inset of Fig. 2, is consistent with the Bragg wavelength of the DFB grating.”*

3 Response to Reviewer 3

Specific comment 1

The authors focus on circular apertures. In their conclusion, they state that by tuning the shape and size of the aperture, further applications can be envisioned for the proposed approach. Did the authors consider using other shapes for the aperture in their current work? If so, can they provide some information to show what would be the result of a different shape on the reflectivity and transmission?

Answer

Thank you for the comment. Besides the circular shape, we also analyzed the elliptical aperture shapes in the revised manuscript. The influence by the aperture shape is discussed in the answer to the Specific comment 5 of Reviewer 1 and the manuscript is revised as follows:

- Line 67: *“In this work, only circular shaped aperture is considered, for the ease of fabrication tools. However, the shape of the aperture is not restricted. We show in Supplemental Material Sec. B that a higher transmission efficiency with nearly the same reflectivity can be achieved by an elliptical aperture.”*
- new section in the Supplemental Material: Sec. B, “Analysis on the aperture shape”

We analyze the reflectivity and the transmissivity dependence on the shape of the aperture using 3D COMSOL simulation. We show that compared to the circular aperture as presented in the main text, a higher transmission efficiency with nearly the same reflectivity can be achieved by an elliptical aperture. We also discover that the optimization of the modal reflectivity is more sensitive to the width of the aperture than the height, in agreement with the polarization feature of the structure as discussed in the manuscript.

Specific comment 2

In supplementary section D, the authors show the characterization of three other devices with the “same material, similar dimensions and coating structures” to prove the reproducibility of their approach. Indeed, the threshold reduction for all of them is a convincing proof. However, there are also significant differences in the extracted power, especially between device 1 and 3. This raises a question regarding how similar these devices are. In fact, I suggest quantifying the differences between these devices so that one can try to distinguish between the effect of different dimensions and different coating structures, for example.

Answer

Thank you for the comment. We added the structural details of the four devices into the Supplementary Material of the revised manuscript. The Sec. D “Additional measurement results of the threshold reducing experiment” of the Supplemental Material is updated as follows:

- Supplemental Material, Sec. D, line 137: *“The structural details of the four devices are listed in Tab. S1 .”*
- Table S1: Structural details of the four devices presented in Fig. S11.

The structural parameters in the added table include the thicknesses of the dielectric and the metallic layers, the aperture diameters, the device lengths and widths, and how are these devices mounted. Note that the alignment between the apertures and the centers of the waveguides also slightly differ from one to another, but cannot be quantitatively compared. The fact of the threshold reducing proves that the alignment in all these four devices are sufficient.

The major reason for the power difference between device I and III is attributed to the difference in the waveguide width. This also explains that the threshold current density of device III is higher than all other lasers. The lasing condition is described

by:

$$\Gamma g_{th} = \alpha_{tot} \quad (1)$$

where Γ is the overlap factor with the active region, g_{th} is the gain at the threshold current density and α_{tot} is the total loss. A narrower waveguide results in a lower Γ , thus requires a higher threshold current density to achieve lasing.

Specific comment 3

The experiments reported in the main text were performed on a coated device with a Al₂O₃ thickness of 700nm. Yet, according to the simulations of fig. 1b, the maximum in the modal reflectivity is at 900nm, which would grant the greater threshold reduction. Therefore, one must conclude that this specific choice of Al₂O₃ thickness was done to maximize the transmission efficiency. The underlying criterium should be made clear in the text, though.

Answer

Thank you for the comment. We revised the manuscript as follows:

- Line 78: *“The Al₂O₃ thickness is selected as 700 nm for an optimal balance in both the enhanced modal reflectivity and the transmission efficiency [Fig. 1(b)]. The Au layer thickness is 200 nm.”*

Specific comment 4

The use of noble metals for the fabrication of plain HR coatings for mid-infrared QCLs is described in the paragraph of line 27. Yet, this solution is not a new one. One can find previous instances in the literature, such as Page, H., P. Collot, A. Rossi, V. Ortiz and C. Sirtori. “High reflectivity metallic mirror coatings for mid-infrared ($\lambda \approx 9 \mu\text{m}$) unipolar semiconductor lasers.” Semiconductor Science and Technology 17 (2002): 1312-1316 for example. A citation is consequently needed.

Answer

Thank you for the comment. The mentioned reference is cited as Ref. 37 in the updated manuscript.

Specific comment 5

There are not many details about the simulations. As a bare minimum, I would like to know if the simulations were all performed in 3D and how the boundary conditions were set. Alternatively, the authors could add a paragraph specifying all relevant information.

Answer

Thank you for the comment. We agree that more details on the simulation should be added. In the updated manuscript, we introduced a new subsection “Simulation details” (line 147) to the “Methods” section of the main text, where we elaborated the details of the 3D COMSOL simulations, including the setting of the boundary conditions:

Line 148: *“The simulated data shown in Figs. 2, 3 are obtained with 3D COMSOL Multiphysics models. The active region is approximated as a rectangular shaped waveguide with the width of 2.8 μm and the height of 2 μm . The cladding InP thickness is set as 3.37 μm . The boundary condition of the top cladding surface is set as perfect electric conductor (PEC). A numeric port is set to the entrance of the waveguide to excite the waveguide mode. Scattering boundary conditions are applied to all other boundaries, including the bottom substrate surface, the top surface of the coating layers and all the sidewalls. Due to the transverse mirror symmetry of the structure, the model is sometimes halved by the symmetry plane and perfect magnetic conductor (PMC) is applied on the cut-plane interface in order to reduce the number of the meshing elements without influencing the simulation results. The choice of PMC boundary condition is due to the TM polarization of the mode. The refractive indices of the Al₂O₃ and the evaporated Au are adopted from the built-in COMSOL library at the target wavelength of 4.5 μm . The refractive indices of InP and the active region are around 3.10 and 3.35, respectively.”*

References

1. Yu, N. *et al.* Quantum cascade lasers with integrated plasmonic antenna-array collimators. *Opt. Express* **16**, 19447–19461 (2008).
2. Yu, N. *et al.* Small-divergence semiconductor lasers by plasmonic collimation. *Nat. Photonics* **2**, 564–570 (2008).
3. Yu, N. & Capasso, F. Wavefront engineering for mid-infrared and terahertz quantum cascade lasers. *JOSA B* **27**, B18–B35 (2010).
4. InfraTec. <http://www.infratec-infrared.com/sensor-division/service-support/glossary/infrared-detector/>.

List of changes

All revised text in both the main text and the Supplemental Material are highlighted by the color of blue.

1. Main text:

- Last line in “Abstract” (line 7)
- Lines 14-17
- Lines 30-31
- Lines 35-39
- Line 41
- Line 45 (corrected two typos)
- Lines 53-54
- Lines 57-59
- Lines 67-69
- Lines 78-79
- Lines 86-87
- Lines 115-116
- Lines 123-126
- Lines 137-138
- new subsection “Simulation details” (lines 147-157) in the “Methods” Section
- Caption of Fig. 1: added one sentence
- new referenced added in the main text: Refs. 28, 29, 30, 37, 38

2. Supplemental Material:

- Merged the previous sections “Optimized aperture diameters at each insulating layer thickness in Fig. 1(b)”, “Cross-sectional field patterns with different aperture dimensions” and a new subsection A.1 “Beam focusing by the metallic aperture” into a new Section A “Additional data and explanation of the reflectivity enhancement”
- new Section B. “Analysis on the aperture shape”
- rewrote Section C. “Additional information on the focused ion beam milling”
- new Tab. S1 into the Section D. “Additional measurement results of the threshold reducing experiment”
- new Section F. “Reflectivity enhancement in TE modes at the wavelength of 1550 nm”
- new Section G. “Transmission enhancement at a large aperture dimension”

REVIEWER COMMENTS

Reviewer #2 (Remarks to the Author):

the authors have substantially answered all my comments and have improved the quality of the manuscript to a level acceptable for publication in Nature Communications.

Reviewer #3 (Remarks to the Author):

The authors addressed all comments and provided satisfactory answers. The article can now be published.

Response to the comments of reviewers

Zhixin Wang^{1,*}, Filippos Kapsalidis¹, Ruijun Wang¹, Mattias Beck¹, and Jérôme Faist^{1,*}

¹ETH Zürich, Institute of Quantum Electronics, Auguste-Piccard-Hof 1, Zürich 8093, Switzerland

*corresponding author: Zhixin Wang (zhixwang@phys.ethz.ch); Jérôme Faist (jfaist@ethz.ch)

Comment from Reviewer 1

The additional theoretical computational work is appreciated, however, the work still feels like it is lacking a more comprehensive demonstration. The results are still lacking in depth beyond a single demonstration and the introduction is still longer than the rest of the text. The authors say, "Quite obviously, the devices presented here as an example are not fully optimized." Then why don't they optimize it to make a better presentation? It is not clear, given the limited footprint of the aperture, whether it even is possible to further optimize the concept and show an improved threshold beyond what they demonstration, which is modest at best.

Answer

Thank you for the comments.

- First of all, we would like to emphasize that the key point of this work is the counter-intuitive physical phenomenon rather than a record-breaking value in the QCL threshold dissipation power. In this manuscript, we theoretically presented and experimentally proved that both the reflectivity and the transmissivity of a metallic coating can be simultaneously enhanced "just" by milling a subwavelength aperture. Indeed, this phenomenon allows a new record of threshold consumption power in QCLs. However, we believe that potential impact of this work is far beyond this record.

In this manuscript, we pointed out it is very crucial to suppress the diffraction loss when pushing the limit of QCL threshold dissipation power. To our knowledge, this is the first time that the diffraction loss is systematically discussed for achieving low-dissipation QCLs. In this regard, our work paves the way towards many other designs for the future work of low-dissipation QCLs. Although these designs are not limited to the presented patterned coating, the control of the diffraction loss should nevertheless be carefully considered in all future works. In addition, as shown in Sec. F of the Supplementary Material, the reflectivity enhancement is also valid for a TE-polarized mode at the wavelength of 1550 nm. Therefore the potential impact of the presented work is not limited to QCLs.

- Secondly, we wrote "the devices presented here as an example are not fully optimized" because the threshold current in this work is limited by the cavity length of the devices and thus by the chip cleaving capabilities. Currently with mechanically cleaving, it is challenging to make QCL cavities much shorter than 250 μm . We are still exploring the technique of defining the waveguide facets via ICP dry-etching, which potentially allows us to reduce the cavity length to much below 200 μm . This is never realized for QCLs before and this technique would enable a much lower threshold with the same achieved modal reflectivity.

As an example to show this, we consider two lasers, laser 1 and laser 2, with differences in only the cavity length and the facet reflectivity. Laser 1 has a 265 μm -long QCL ridge with 95.6% reflectivity on both facets. It is essentially the presented device before any patterning in the coatings (blue curve in the Fig. 2 of the main text). Laser 2 has a QCL ridge length of 143 μm by dry-etching and 97.6% reflectivity on both sides, which equals to the reflectivity achieved with the patterned metallic coatings. According to $\alpha_m = |\ln(R^2)|/2L$, laser 1 and laser 2 have the same mirror loss. Therefore, we expect laser 2 to share the same threshold current density as laser 1 (2.22 kA/cm^2). In this case, the threshold current

of laser 2 will be 7.8 mA at 20°C, 30% lower than achieved record in this manuscript (green curve in Fig. 3 of the main text).

In principle, the dry-etching technique could allow ridge lengths down to below 100 μm , and we expect even lower threshold dissipation power with such devices, keeping the same value of reflectivity realized in this work. For example, a QCL with the length of 50 μm and 97.6% reflectivity would have a mirror loss of 4.9 cm^{-1} . According to our measured relationship between the current density and the gain, the threshold current density in this case will be increased to 3.8 kA/cm^2 , leading to a threshold current of 5 mA, which is less than one half of the achieved record in this work. Therefore, we believe that the technique of defining the ridge facets via dry-etching would potentially lead to very promising devices with a much lower threshold dissipation power in the future.

- The manuscript is updated as follows:
 - Line 18: paragraphs of the section “Introduction” moved into a new section “Theoretical simulations”
 - Line 114-115: “*We demonstrated that, in general, it is crucial to control the diffraction loss for achieving a low dissipation QCL. In the future, there is still room for further optimization. For example, using facets realized via dry etching techniques would enable much shorter cavities to be fabricated and consequently allow a lower threshold dissipation power while keeping the same reflectivity realized in this work*”.
 - Supplemental Material: Figure S8 re-plotted into colored contour images for a better vision. The data is unchanged.